# Estimating the extrinsic incubation period of malaria using a mechanistic model of sporogony

**Isaac J. Stopard** *, **Thomas S. Churcher**, **Ben Lambert** ¤

MRC Centre for Global Infectious Disease Analysis, School of Public Health, Faculty of Medicine, Imperial College London, London, United Kingdom

¤ Current address: Department of Computer Science, University of Oxford, Oxford, United Kingdom
* isaac.stopard11@imperial.ac.uk

**Data Availability Statement:** All data are held on a public GitHub repository - https://github.com/IsaacStopard/mSOS.

**Funding:** IJS would like to thank the Natural Environment Research Council (NE/P012345/1) for

## Abstract

During sporogony, malaria-causing parasites infect a mosquito, reproduce and migrate to the mosquito salivary glands where they can be transmitted the next time blood feeding occurs. The time required for sporogony, known as the extrinsic incubation period (EIP), is an important determinant of malaria transmission intensity. The EIP is typically estimated as the time for a given percentile, $x$, of infected mosquitoes to develop salivary gland sporozoites (the infectious parasite life stage), which is denoted by $EIP_x$. Many mechanisms, however, affect the observed sporozoite prevalence including the human-to-mosquito transmission probability and possibly differences in mosquito mortality according to infection status. To account for these various mechanisms, we present a mechanistic mathematical model, which explicitly models key processes at the parasite, mosquito and observational scales. Fitting this model to experimental data, we find greater variation in the EIP than previously thought: we estimated the range between $EIP_{10}$ and $EIP_{90}$ (at 27˚C) as 4.5 days compared to 0.9 days using existing statistical methods. This pattern holds over the range of study temperatures included in the dataset. Increasing temperature from 21˚C to 34˚C decreased the $EIP_{50}$ from 16.1 to 8.8 days. Our work highlights the importance of mechanistic modelling of sporogony to (1) improve estimates of malaria transmission under different environmental conditions or disease control programs and (2) evaluate novel interventions that target the mosquito life stages of the parasite.

## Author summary

*Anopheles* mosquitoes become infected with malaria-causing parasites when blood feeding on an infectious host. The parasites then reproduce via a number of life stages, which begin in the mosquito gut and end in the salivary glands, where the newly formed infectious parasites can be transmitted to another host the next time a mosquito blood feeds. This delay in the mosquito becoming infectious, known as the extrinsic incubation period (EIP), is long relative to mosquito life expectancy. Consequently, the EIP is important in determining whether a mosquito is able to transmit malaria. The EIP is typically estimated

PhD funding administered through the Quantitative Methods in Ecology and Evolution Centre for Doctoral Training. IJS, TSC & BL would like to acknowledge the MRC Centre for Global Infectious Disease Analysis (grant reference: MR/R015600/1), this award is jointly funded by the UK Medical Research Council (MRC) and the UK Department for International Development (DFID) under the MRC/DFID Concordat agreement and is also part of the EDCTP2 programme supported by the European Union. The funders had no role in study design, data collection and analysis, decision to publish, or preparation of the manuscript.

**Competing interests:** The authors have declared that no competing interests exist.

by fitting a statistical model to parasite data from the dissection of numerous mosquitoes. The large variability in development times and parasite numbers that exists between parasites, mosquitoes and environments means that estimating the EIP is difficult. Here, we introduce a mathematical model of the population dynamics of the mosquito life stages of the parasite, which mimics key characteristics of the biology. We show that the model's parameters can be fit so that its predictions correspond with experimental observations. Our work is a step towards a realistic model of within-mosquito parasite dynamics, which can be applied to help understand heterogeneity in malaria transmission.

## Introduction

Malaria remains a leading cause of morbidity and mortality worldwide, with an extremely inequitable distribution: over 400,000 people, primarily children under the age of five in sub-Saharan Africa, die annually due to malaria [1]. The widespread use of vector control tools that kill adult *Anopheles* mosquitoes is largely responsible for a historical decline in malaria incidence [2]; a result foretold by early mathematical models, which predicted the sensitivity of malaria transmission to adult mosquito survival [3,4]. For a newly infected mosquito to become infectious, it must survive the extrinsic incubation period (EIP). Since the EIP is long relative to mosquito life expectancy, only older mosquitoes can pass on infection meaning malaria transmission responds acutely to changes in survival [3,5].

The EIP is defined as the duration of sporogony: the obligate reproduction of malaria-causing *Plasmodium* parasites (henceforth parasites) within the mosquito [6]. First, female mosquitoes feed on an infectious host. A proportion of these mosquitoes, as determined by the human-to-mosquito transmission probability [5], ingest male and female *Plasmodium* gametocytes within the red blood cells (RBCs) of the blood-meal [7]. The change of parasite host (from human to mosquito) involves certain environmental changes, including a decrease in temperature, which collectively trigger gametogenesis and the parasites to emerge from the RBCs [8,9]. Fertilisation occurs within the mosquito midgut, where gametes fuse into a single zygote, which differentiates into a motile ookinete [10]. Within a few days, ookinetes migrate across the mosquito midgut epithelial wall, and the parasites that survive the mosquito innate immune response [11] go on to form immobile oocysts beneath the midgut basal lamina [12]. The number of oocysts remains relatively constant as they grow in size and their genome mitotically replicates [12]. Oocysts then burst with each releasing hundreds of infectious sporozoites, which migrate to the mosquito salivary glands, completing the EIP [6,13].

The EIP and human-to-mosquito transmission probability are typically estimated in the laboratory using experimentally introduced infections. Laboratory reared mosquitoes are fed on infectious blood through a membrane feeder, and the parasites are allowed to develop before the mosquitoes are dissected to determine the presence and number of oocysts or sporozoites [13,14]. Since a mosquito can be dissected only once, it is not possible to observe parasites dynamics within a single mosquito (but see [15] for a novel approach to estimate this). Numerous dissections are therefore used to reconstruct the temporal dynamics of sporogony in the population at large [6].

Historically, the EIP has been recorded as the time from blood feeding until the first mosquito is observed with any salivary gland sporozoites [16,17]. Parasite development rates are, however, highly heterogenous due to differences in mosquito nutrition [18,19] and environmental temperature [14,20], and considerable variation remains even after accounting for these factors [14,18]. Mosquito and parasite genetic differences may also contribute to

variation [16], but even in genetically identical cells reaction rates are noisy [21]. Given this variation, any single measure–especially one based on a single mosquito, which has historically been the case–does not adequately represent the EIP. Instead, more recent studies attempt to characterise the variability in the EIP, which is reported as the time for given percentiles of infected mosquitoes to display sporozoites [16]. A rigorous framework for characterising the distribution of the EIP is yet to be defined.

Not all mosquitoes fed infectious blood will develop observable oocysts or sporozoites. The human-to-mosquito transmission probability depends on many factors including differences in blood meal size, gametocyte density [7,22,23], the mosquito immune response [11,24] and midgut microbiota [25,26] amongst others. To estimate the EIP percentiles, it is necessary to determine the number of infected mosquitoes within the sample as a proportion of all infected individuals. To do this, it is assumed that the maximum observed oocyst- or sporozoite- prevalence is the actual proportion of mosquitoes with viable infection [16]. Focussing on raw temporal changes in observed prevalence without an underlying mechanistic model may, however, overlook key processes. In the laboratory, malaria infections may alter mosquito survival in infected mosquitoes [27], meaning the observed maximum prevalence may not represent the true proportion of infected mosquitoes. What's more, certain variables can impact multiple mechanisms simultaneously [28], requiring a finer-scale understanding of the constituent processes.

Temperature, for example, can modulate the parasite development rate [14,20], survival of laboratory mosquitoes [14] and mosquito immune response [29], causing differences in vector competence [30]. To date, the relationship between temperature and the EIP has been modelled using a degree-day model, which parameterises the total amount of heat required to complete sporogony [31], or existing functions (quadratic or Brière) [32,33]. Parameterisations of both these methods rely on a point estimate of the EIP and do not model the constituent processes that produce these observations. Mechanistic models that simulate parasite population dynamics during sporogony may provide a more informative framework [34–36].

Here, we introduce a mathematical model, the "multiscale Stochastic model Of Sporogony (mSOS)", to capture the temporal dynamics of *Plasmodium* sporogony. By explicitly modelling the underlying biology, at the parasite, mosquito and experimental scales, we aim to provide an accurate representation of parasite development [37]. We fit mSOS to laboratory data and present new EIP estimates which contrast with those estimated using the predominant approach in the literature. This is intended to illustrate the application and implications of using a mechanistic model of sporogony, and not to summarise current understanding of the EIP, which would require a wider systematic review. We also demonstrate how mSOS can be used to estimate the impact of temperature on *Plasmodium falciparum* sporogony within *Anopheles* mosquitoes. All code and data needed to recapitulate our results are available at https://github.com/IsaacStopard/mSOS.

## Models and methods

### Data

We are unaware of a single study with sufficient data to fit the model. To capture the full population dynamics of oocysts and sporozoites, we combined data from four published studies (Table 1). Most data came from Standard Membrane Feeding Assays (SMFA) from the same laboratory, conducted with the same parasite-vector combination [14,19,38]. These studies collected data over a range of days and temperatures using a laboratory strain of *P. falciparum* (NF54) and laboratory-reared *Anopheles stephensi* mosquitoes. To examine oocyst prevalence prior to day five, we also included data from a Direct Membrane Feeding Assay (DMFA) [39],

**Table 1. Summary of different studies, parasite-vector combinations and experimental proceedures used to parameterise the model.**

| Malaria species (strain) | Vector species (strain) | Days post blood feeding when oocyst presence or load was determined by dissection | Days post blood feeding when sporozoite presence was determined by dissection | Dead mosquitoes counted? | Adult mosquito housing temperature (°C) | Reference |
|---|---|---|---|---|---|---|
| *P. falciparum* (NF54) | *A. stephensi* (laboratory) | 7 (oocyst load) | 15 | Yes, infectious blood fed & uninfected blood fed (control) mosquitoes | 27, 30 & 33 | Murdock et al. [38] |
| *P. falciparum* (NF54) | *A. stephensi* (laboratory) | Not recorded | Varied with temperature: 10 to 23 & 25 (21°C), 9 to 23 & 25 (24°C), 8 to 18, 20, 22, 24 & 25 (27°C), 6 to 17 & 19 (30°C), 5 to 15, 17 & 19 (32°C) and 5 to 14 & 16 (34°C). | Yes, infectious blood fed mosquitoes only | 21, 24, 27, 30, 32 & 34 | Shapiro et al. [14] |
| *P. falciparum* (NF54) | *A. stephensi* (laboratory) | 5, 6, 7, 8, 9 & 10 (oocyst load) | 9, 10, 11, 12, 13, 14, 15 & 16 | Yes, infectious blood fed mosquitoes only | 27 | Shapiro et al. [19] |
| *P. falciparum* (wild) | *A. gambiae s.s.* (laboratory) | 3, 4, 5, 6, 7 & 8 (oocyst presence used) | Not used | No | 27–28 | Bompard et al. [39] |

which used wild *P. falciparum* parasites to infect laboratory-reared *Anopheles gambiae sensu stricto* mosquitoes. It is assumed there is no difference between the two parasite-vector combinations. All studies replicated the following experimental protocol: mosquitoes were raised from larvae under laboratory conditions; adult female mosquitoes were then fed on *P. falciparum*-infected blood via a membrane feeder; at various days post infection, blood fed mosquitoes were collected, dissected and the presence of oocysts or salivary-gland sporozoites identified by microscopy. The oocyst load within each mosquito was estimated by counting observable oocysts. Sporozoites were quantified by their observed presence or absence during dissection. Only mosquito larvae kept on a high food diet were used in these analyses, and, in all cases, mosquitoes were housed at a constant temperature and humidity (see Table 1 for the range of different temperatures explored).

## Model overview

mSOS models the oocyst and salivary gland sporozoite (henceforth sporozoite) life stages, as oocysts are the most widely recorded outcome of membrane-feeding assays, and sporozoites are the most epidemiologically important life stage. We explicitly model the counts of oocyst- and sporozoite-positive mosquitoes to determine how the prevalence (i.e. the percent of positive mosquitoes among the sample) of each life stage varies over time. We also model oocyst intensity (mean oocyst load among the sample), as it provides more information on the underlying parasite dynamics and is considered a more reliable estimate of human-to-mosquito transmission [40,41].

Sporogony is modelled at multiple scales, and all parameters are fit simultaneously in a single model. At the parasite scale, we model the sequence of three life stages: inoculant  oocyst  sporozoite (Fig 1A). We do not model the pre-oocyst life stages since they are harder to count and are rarely recorded. Rather, these are represented in a notional "inoculant" stage ingested during blood feeding, each of which develops into a single oocyst. mSOS captures both the time it takes for each oocyst to appear and the time for each oocyst to develop into sporozoites. The time taken for each of these transitions to occur is assumed independent of the other, and we model these times stochastically.

At the mosquito scale, mosquitoes are initially infected with different numbers of inoculant stage parasites, with each load determined stochastically. We also allow infectious blood fed mosquitoes to avoid infection (via a human-to-mosquito transmission probability) (Fig 1B).

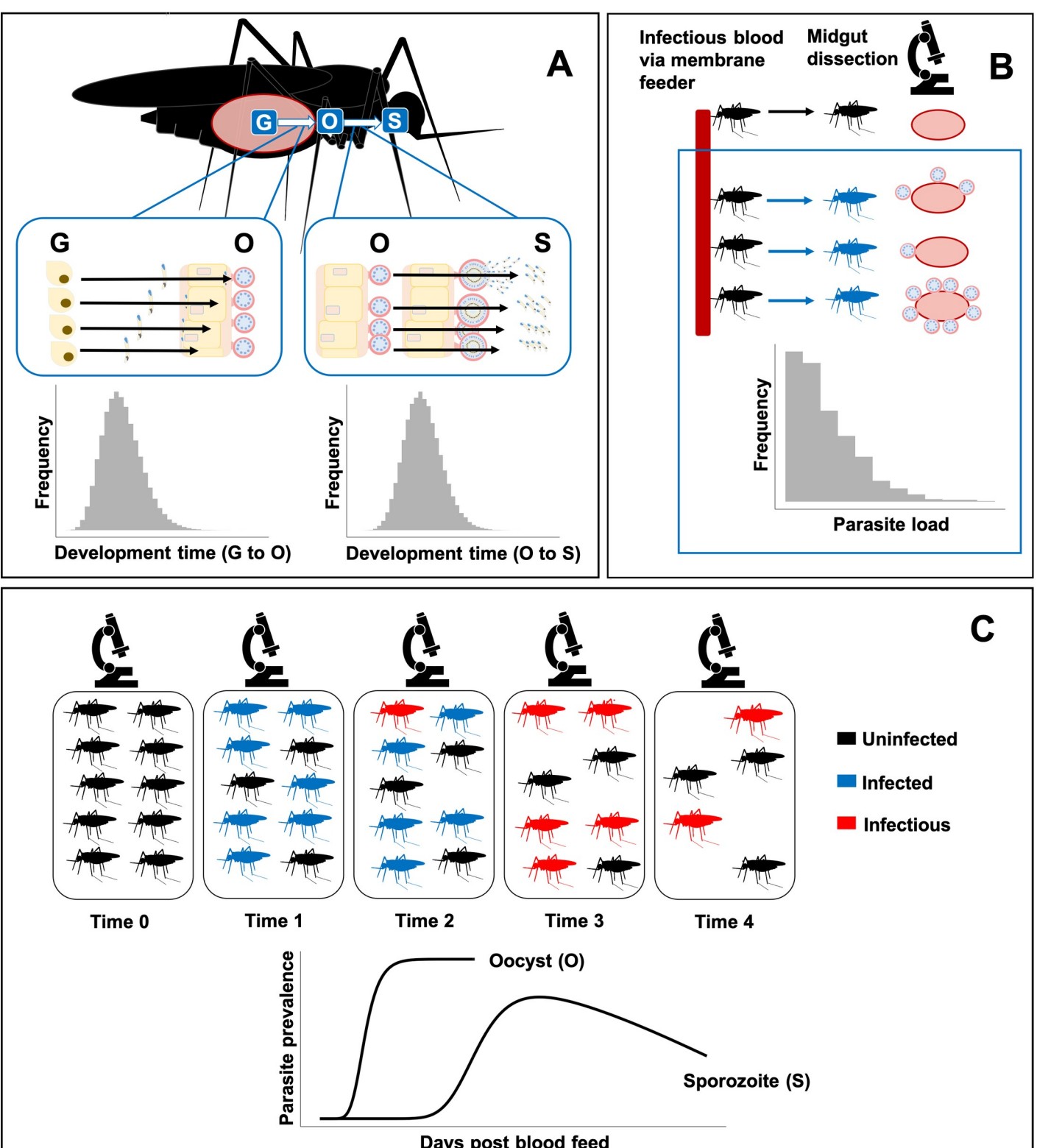

**Fig 1. Structure of mSOS: a multiscale model of the population dynamics of *Plasmodium falciparum* during sporogony.** (A) Multiple malaria parasites are found within a single mosquito; we separately model development time from inoculation at blood feeding (G) to oocyst (O) and from oocyst to salivary gland sporozoites (S). If dissected, a mosquito is "positive" for a particular parasite life stage if at least a single parasite has developed. We do not model the decline in observed oocyst numbers due to oocyst bursting, since we do not have sufficient later oocyst observations. (B) Mosquitoes are infected via a membrane feeder; parasite load varies in

each due to differences in the number of parasites ingested and variation in mosquito immune response. (C) Temporal dynamics of sporozoite prevalence within a mosquito population: following the infectious blood feed, a proportion of the population is infected with malaria parasites. The parasites develop into sporozoites causing the mosquitoes to become infectious. Throughout the experiments, mosquito mortality (in the laboratory) may be greater in infected mosquitoes, resulting in an eventual decline in observed sporozoite prevalence.

Potential differences in survival of the mosquito population due to differences in malaria infection [27] are explicitly modelled.

At the observation level, to reconstruct the temporal dynamics of sporogony at a population level, we recreate populations of parasites nested within a population of mosquito vectors, which then form samples of dissected mosquitoes (Fig 1C). We allow the composition of each sample of dissected mosquitoes to vary according to the proportions of infected and uninfected mosquitoes alive at the time of dissection.

Experimental evidence indicates that the pre-oocyst life stages are the most temperature sensitive [42,43], and temperature affects both the human-to-mosquito transmission probability [28] and laboratory mosquito survival [14]. We therefore allowed these processes to be influenced by temperature. The development time from oocyst to sporozoites and the mean parasite load among infected mosquitoes was assumed to be independent of temperature.

In what follows, we describe the model in detail. To provide greater clarity, a Mathematica file that steps through the derivations is available (S1 Code).

**Parasite scale.** We model the time taken, $T_{GO}$, for each undeveloped oocyst at the time of blood feeding (the notional inoculant stage denoted '$G$') to develop into an observable oocyst ('$O$'). We then model the time taken, $T_{OS}$, for each oocyst to develop, burst, and for the sporozoites to migrate to the salivary glands (we term sporozoites in the salivary glands '$S$'). So, these transitions follow:

$$G \xrightarrow{T_{GO}} O \xrightarrow{T_{OS}} S$$

These development times are modelled stochastically as: $T_{GO} \overset{i.i.d.}{\sim} \Gamma(\alpha_{GO}, \beta_{GO})$ and $T_{OS} \overset{i.i.d.}{\sim} \Gamma(\alpha_{OS}, \beta_{OS})$; that is, we assume that the time it takes a subsequent oocyst to burst into sporozoites is independent of the initial time taken for the oocyst to develop. This assumption also implies the transitions of each parasite occur independently, meaning density dependent population dynamics do not occur. We assume a parameterisation of the gamma distribution such that its mean is $E(T_i) = \alpha_i/\beta_i$, where $i \in (GO, OS)$, $\alpha_i$ is the shape parameter and $\beta_i$ is the rate parameter.

The time required for $G$ to develop into $S$ is given by the sum: $T_{GS} = T_{GO} + T_{OS}$. An analytic form for the cumulative density function (CDF) of the sum of two gamma distributed random variables with different rate ($\beta$) parameters is currently not known. This aggregate distribution can, however, be approximated by a gamma distribution where the mean, $\lambda$, and variance, $\sigma^2$, match that of the true distribution [44],

$$T_{GS} \sim \Gamma(\alpha_{GS}, \beta_{GS}), \tag{2.01}$$

where,

$$\alpha_{GS} = \frac{\lambda^2}{\sigma^2}, \beta_{GS} = \frac{\lambda}{\sigma^2}, \lambda = \frac{\alpha_{OS}\beta_{GO} + \alpha_{GO}\beta_{OS}}{\beta_{GO}\beta_{OS}}, \sigma^2 = \frac{\alpha_{OS}\beta_{GO}^2 + \alpha_{GO}\beta_{OS}^2}{\beta_{GO}^2\beta_{OS}^2}. \tag{2.02}$$

**Mosquito scale.** Mosquitoes are treated as self-contained populations of parasites. When feeding on infectious blood, mosquitoes receive a heterogeneous load of gametocytes, with some receiving none at all [7]. Additionally, not all parasites develop into the observed oocyst

stage due to the innate immune response of some mosquitoes [13,40]. Evidence indicates that, assuming the mosquito is still alive, the majority of oocysts will produce sporozoites [45]. Hence, we term a mosquito "successfully infected" if at least one viable oocyst could develop from the initial infective load given sufficient time and define this probability of successful infection (i.e. the human-to-mosquito transmission probability) as $\delta$. Among successfully infected mosquitoes, the initial load of G-stage parasites, $n$, is modelled by a zero-truncated negative binomial distribution,

$$n \sim NB(\mu, k), n \in [1, \infty), \tag{2.03}$$

where, $\mu$ is the mean and $k$ is the overdispersion parameter. Evidence indicates that $\mu$ and $\delta$ may be jointly influenced by gametocyte density, which is modulated by temperature [30]. Here, however, we treated these parameters as independent, as there was an insufficient combination of gametocyte density and temperature treatments in the data.

To model the observed oocyst or sporozoite presence, we assume a mosquito is measured as "positive" if at least a single parasite has developed to the given life stage. That is, we assume that dissection always uncovers some parasites of a given life stage should any exist.

Let the time taken for parasite $j$ to develop into a subsequent stage be $T_j$ for $j = 1,2,\ldots,n$ parasites within a specific mosquito. Whether a mosquito is positive for a particular stage at time $t$ then depends on whether $t \geq T_{1:n}^{min} = min(T_1, T_2, \ldots, T_n)$. Since we are concerned with the minimum of a series of random variables, we are in the realm of order statistics–see, for example, [46]. Specifically, $T_{1:n}^{min}$ is the 1st order statistic of the sample of $n$ development times. Considering a single parasite, $j$, the probability that it has developed by time $t$ is given by $Pr(t \geq T_j) = Q(\alpha, 0, \beta t)$, where $Q(\alpha, 0, \beta t)$ is the CDF of the gamma distribution governing that particular life stage transition. Specifically, $Q$ is the generalized regularized incomplete gamma function $Q(a, z_0, z_1) = \int_{z_0}^{z_1} t^{a-1}e^{-t}dt / \int_0^{\infty} t^{a-1}e^{-t}dt$, and $\alpha, \beta$ are the shape and rate parameters of the underlying gamma distribution (these parameters will, in general, be different for $T_{GO}$ and $T_{GS}$). Considering $n$ parasites within a single mosquito, the probability that at least one of them has developed by time $t$ is:

$$Pr(t \geq T_{1:n}^{min}) = 1 - \prod_{j=1}^{n} Pr(t < T_j) = 1 - (1 - Q(\alpha, 0, \beta t))^n. \tag{2.04}$$

To model the number of oocysts at a given time within an individual infected mosquito, $Y(t)$, we track the number of G-stage parasites, from the initial load $n$, that have developed by a given time, which follows a binomial distribution,

$$Y(t) \sim B(n, Q(\alpha_{GO}, 0, \beta_{GO}t)), \tag{2.05}$$

where $Q(\alpha_{GO}, 0, \beta_{GO}t)$ is the cumulative probability that an individual parasite has developed into an oocyst by time $t$.

Mosquitoes die throughout the course of the experiments, which we explicitly model using a Cox regression model. We estimate the probability a mosquito is alive immediately prior to time $t$, $A(t)$, by fitting this model to the mosquito survival data (Table 1) and allow for differences in the survival due to infection by comparing groups of mosquitoes exposed to uninfected and infected blood and allowing the hazard to vary with infection status. Mosquitoes exposed to infected blood may not actually become infected, so this group consists of a mix of infected and uninfected individuals. By fitting the complete model, we account for this to estimate infection-specific differences in mortality (see S1 Text for full details). A previous analysis of a subset of the data determined that the Gompertz distribution, in which the mosquito

mortality rate increases with age (senescence) [14], was the best fit, and we use this distribution here.

**Observation level.** During the experiments, samples of mosquitoes are dissected at particular time intervals. From these samples, two possible measurements are recorded: (1) the aggregate count of parasite-positive mosquitoes (either oocysts or sporozoites) and (2) the number of oocysts counted in each of the dissected mosquitoes.

We model the aggregate count of positive mosquito data thus. At each time, a sample of mosquitoes, $D(t)$, are dissected, likely comprising a mix of infected and uninfected individuals. The number of infected mosquitoes within the sample, $I(t)$, is modelled as a binomial random variable,

$$I(t) \sim B(D(t), \delta R(t)),\tag{2.06}$$

where $\delta$ is the probability successful infection occurs during the infectious blood feeding, and $R(t)$ is the fraction of infected mosquitoes alive. $R(t)$ is the ratio of survival probabilities for infected ($E$) and uninfected ($U$) mosquitoes:

$$R(t) = \frac{A_E(t)}{A_U(t)}.\tag{2.07}$$

Note, we assume that once a mosquito is infected it will remain so for the duration of its lifespan [47].

A mosquito may be successfully infected but no parasites observed during dissection if insufficient time has passed since blood feeding. If $n$ was known for each dissected mosquito, the probability dissection would detect parasites of a given life stage, at time $t$, is given by Eq (2.04). In reality, $n$ is not known. We model this uncertainty using Eq (2.03) and incorporate it into the probability any infected mosquito has observable parasites by marginalising $n$ out of the joint distribution,

$$
\begin{aligned}
Pr(t \geq T^{min}) &= \sum_n Pr(t \geq T_{1:n}^{min}|n)Pr(n) \\
&= E_n(Pr(t \geq T_{1:n}^{min}|n)) \\
&= E_n(1 - (1 - Q(\alpha, 0, \beta t))^n) \\
&= \frac{1 - k^k(k + \mu Q(a, 0, \beta t))^{-k}}{1 - \left(\frac{k}{k+\mu}\right)^k}
\end{aligned}
\tag{2.08}
$$

where $T^{min}$ is the time when first parasite of the given life stage appears ($O$ or $S$), and $\mathbb{E}_n(.)$ denotes the expectation with respect to the distribution of $n$ (Eq (2.03)).

The count of infected mosquitoes in which the parasite life stage of interest ($O$ or $S$) is observed, $X(t)$, given a sample of infected mosquitoes, $I(t)$, is then also binomially distributed,

$$X(t) \sim B(I(t), Pr(t \geq T^{min})).\tag{2.09}$$

Within a given sample of mosquitoes, the number of successfully infected mosquitoes, $I(t)$, is not known–we only observe the count of observed infected individuals, $X(t)$, and the total number dissected, $D(t)$. We incorporate this uncertainty in $I(t)$ by marginalising this quantity out of the joint distribution defined in Eqs (2.06) and (2.09), resulting in the following

binomial distribution describing the observed count,

$$X(t) \sim B(D(t), \delta R(t) Pr(t \geq T^{min})).$$ (2.10)

We next detail how our model describes oocyst counts, $Y(t)$. A particular mosquito may yield a zero count when dissected either if it was not successfully infected or if insufficient time has elapsed for any oocysts to develop. The probability a sampled individual is successfully infected is $\delta R(t)$. The count of oocysts to have developed by a given time within a successfully infected mosquito is described by Eq (2.05): this probability distribution depends on $n$ (the number of G-stage parasites)–an unknown quantity. To derive the sampling distribution of oocyst counts, we marginalize $n$ out of the joint distribution assuming its uncertainty is described by Eq (2.03). Combining this with our underlying uncertainty regarding the mosquito's infection status results in the following expression describing the probability of counting $Y$ oocysts at time $t$, in a dissected mosquito,

$$Pr(Y(t)|Q_{GO}(t), \delta, R(t), t, \mu, k)$$

$$= \begin{bmatrix} (1 - \delta R(t)) + \delta R(t) \left( \dfrac{\left(\dfrac{k}{k+\mu}\right)^k - \left(\dfrac{k}{k+Q_{GO}(t)\mu}\right)^k}{-1 + \left(\dfrac{k}{k+\mu}\right)^k} \right), \text{ if } Y = 0 \\[2em] \delta R(t) \dfrac{\left( k^k (Q_{GO}(t)\mu)^Y (k + Q_{GO}(t)\mu)^{(-Y-k)} \begin{pmatrix} -1+x+k \\ -1+x \end{pmatrix} \right)}{1 - \left(\dfrac{k}{k+\mu}\right)^k}, \text{ if } Y \geq 1. \end{bmatrix}$$ (2.11)

where, for notational convenience, we term $Q_{GO}(t) = Q(\alpha_{GO}, 0, \beta_{GO}t)$. From Eq (2.11), the mean oocyst intensity within the population is given by $\delta R(t) \frac{Q_{GO}(t)\mu}{1-\left(\frac{k}{k+\mu}\right)^k}$. Mosquitoes are typically dissected for oocysts before any burst, so we do not model how bursting would lead to a decline in oocyst prevalence or intensity.

## Incorporating temperature-dependence

First, we fit the model to data collected under standard insectary conditions (27°C), which included the complete range of oocyst intensity, oocyst prevalence, sporozoite prevalence and survival data (grouped according to whether mosquitoes were infectious blood fed or control). Next, to investigate the impact of temperature on the EIP, we fit the model to the data collected at each other temperature, which included sporozoite prevalence and survival data only. Due to the lack of oocyst data, development times from day of blood feeding to sporozoite were estimated as a single gamma distribution with shape, $\alpha_{GS}$, and rate, $\beta_{GS}$. Data collected at 33°C was excluded as mosquitoes were only sampled on a single day. We refer to these individual temperature model fits collectively as the "*single temperature* models".

To estimate a functional relationship between temperature and EIP, we fit the model to all the data simultaneously (the "*all temperature* model"). To do so, we first plotted the $\alpha_{GS}$ and $\beta_{GS}$ parameters of the *single temperature* models against temperature (S1 Fig). This indicated an approximate linear relationship between temperature and the development rate, $\beta_{GS}$. Laboratory studies have found the early part of sporogony of laboratory strains of *P. falciparum*, in *A. stephensi*, is most temperature sensitive [42,43], so we assumed the temperature variation

affected the rate at which parasites develop from $G$ to $O$. The development rate was then modelled as a linear function of temperature, $C$,

$$\beta_{GO} = m_\beta C + c_\beta. \tag{2.12}$$

In the laboratory, the infection of mosquitoes has been found to decline at high temperatures, due to factors such as the impact of temperature on vector immunity [28] and the establishment of gametocytes [30]. To account for this, we included a relationship between temperature and the human-to-mosquito transmission probability, δ, modelled as a logit-linear function,

$$\delta = \frac{1}{1 + e^{-(m_\delta C + c_\delta + \rho_i)}}. \tag{2.13}$$

To capture differences between membrane feeding assays [40], we included a hierarchical term, $\rho_i$, in Eq (2.13) where,

$$\rho_i \sim N(0, \sigma_\delta), \tag{2.14}$$

and $i$ refers to an individual experiment, defined as any unique combination of study and temperature treatments (i.e. the combination of reference and adult mosquito housing temperature values in Table 1).

Kaplan-Meier plots of mosquito survivorship and existing studies indicated that laboratory *A. stephensi* survival depended on temperature (S2 Fig). To account for this, a temperature-dependent mortality term was incorporated into the Cox model hazard,

$$h_0(t)e^{(\beta_C C + \beta_E E + \varepsilon_i)}, \tag{2.15}$$

where C is temperature, E is the infection status ($E = 1$ indicates successful infection; and $E = 0$ for uninfected individuals), and $h_0(t)$ is the baseline hazard modelled by a Gompertz distribution. To account for experimental heterogeneity in mosquito survival between experiments, a hierarchical error term, $\varepsilon_i$, was included in Eq (2.15) where,

$$\varepsilon_i \sim N(0, \sigma_{survival}). \tag{2.16}$$

To facilitate model fitting, we scaled the temperature values such that temperature was centred around 0 with a standard deviation of 1.

## Model fitting

The models were fit under a Bayesian framework, using the probabilistic programming language Stan (version 2.21.0 with R version 3.6.3) [48], which implements the No-U-turn Markov chain Monte Carlo (MCMC) sampler [49]. We specified weakly informative priors following a literature search (S1 Table). For each model fit, four chains were run, each with 1500 warmup iterations and 4500 iterations in total. Convergence was determined by $\hat{R} <$ 1.01 for all parameters, and bulk and tail effective sample sizes (ESS; an estimate of sampling efficiency) greater than 400 (see S2 Table for these values). We visually examined the trace plots (S3 Fig), pairwise plots of the MCMC parameter values (S4 Fig) and difference between prior and posterior distributions (S5 Fig).

## Model investigation

To compare our estimates with the predominant approach, we estimated the EIP and human-to-mosquito transmission probability using a logistic model [14,16,19,28,50], which we fit using non-linear least squares (see S1 Text). Given a decline in the observed sporozoite

prevalence at later time points, the data were subset to include only data points before the time of peak observed sporozoite prevalence. This method has previously been used to enable the logistic function to be fit to modal data [14], but requires the following assumptions to be true: (1) the time when the observed sporozoite peak occurs in those mosquitoes sampled reflects the same attribute in the underlying population, and (2) all mosquitoes have developed sporozoites by the time observed sporozoite prevalence starts to decline.

Data did not exist to investigate the impact of parasite load experimentally. We therefore used our fitted model to simulate the temporal dynamics of sporogony with different mean parasite loads (among infected mosquitoes) holding all other parameters at their mean posterior values for 27˚C ("as a" sensitivity analysis).

## Results

### Temporal dynamics of sporogony

We first considered the observed temporal dynamics of sporogony at a single temperature (27˚C; standard insectary conditions). All central values we report are the posterior predictive means and the credible intervals are the 95% central posterior estimates; times reported are the number of days post infectious blood feed. From the model fit, we estimate that oocyst prevalence was 10% at 2.2 days (CI: 2.1–2.3 days) and peaked at 69% (CI: 68–71%) after approximately 5.9 days (CI: 5.8–6.0 days) (Fig 2A). The modelled oocyst intensity peaked at an average of 2.0 (CI: 1.9–2.1) oocysts per mosquito, shortly after oocyst prevalence peaked (Fig 2B). In the raw data, sporozoite-positive mosquitoes were first observed on day 10 when the sporozoite prevalence was 1.4%; our model estimates the appearance of sporozoite-positive mosquitoes earlier, with the model estimating a sporozoite prevalence of 5% at 9.0 days (CI: 8.8–9.2 days) (Fig 2C). The time required for sporozoites to appear varied between mosquitoes, and, only after 15.9 days (CI: 14.2–16.0 days) did the modelled sporozoite prevalence peak at 63% (CI: 61–65%).

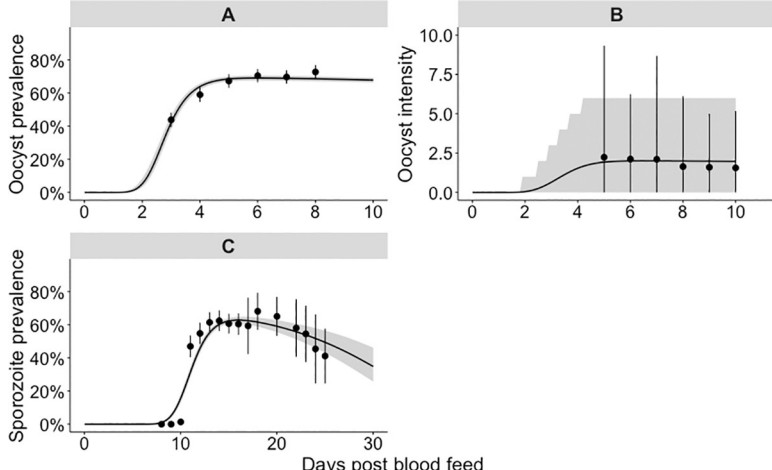

**Fig 2. Single temperature 27˚C model fit to the oocyst and sporozoite data.** The panels show our model fit to the 27˚C dataset: panel A to the oocyst prevalence, panel B to the oocyst intensity data and panel C to the sporozoite prevalence. (A & C) points show parasite prevalence of the laboratory mosquito data (95% confidence intervals are given by the point range). The grey shaded area represents the 95% credible interval of the model posterior predictive means, the median posterior predictive mean is shown by the black line. (B) The points show the mean parasite load among all blood fed mosquitoes (intensity); the point range indicates the 2.5%–97.5% quantiles of the raw data. The shaded area represents the 2.5%–97.5% quantiles of the negative binomial distribution; where the location and overdispersion parameters are set to their posterior means.

At the individual parasite level, we consider the time taken for each modelled parasite life stage transition to occur: the mean time taken was 3.4 days (CI: 3.3–3.6 days) for G to O, 9.2 days (CI: 8.9–9.5 days) for O to S and 12.6 days (CI: 12.3–12.9 days) for G to S (S6A Fig). For G to O, the majority of individual parasites (which we took as 99.5% of all parasites) had undergone each of these transitions at 6.1 days (CI: 5.6–6.6 days); for O to S, at 14.7 days (CI: 13.9–15.5 days); and, for G to S, at 18.5 days (CI: 17.8–19.3 days) (S6B Fig).

## Impact of temperature

We next considered the impact of temperature on sporogony. The *single temperature* models fitted the sporozoite prevalence data well across the entire range of temperatures (S7 Fig). The *all temperature* model fitted the sporozoite prevalence data well at lower temperatures (between 21˚C and 30˚C); at higher temperatures, the model tended to overstate the EIP (Fig 3). The survival of both infectious blood fed and control mosquitoes was only available at 27˚C, 30˚C and 33˚C, and consequently the *single temperature* model survival parameters had greater freedom to vary. Indeed, the *all temperature* model (S8 Fig) fits to the survival data were less variable and more predictable across different temperatures than the *single temperature* models (S9 Fig). Consequently, here we provide the *all temperature* model (Figs 3 and S10) results since they are less likely to overfit the data.

For the *all temperature* model, infection resulted in a higher risk of mosquito death, with a hazard odds ratio of 1.65 (CI: 1.47–1.86). Increases in temperature elevated mosquito mortality with a hazard odds ratio of 1.57 (CI: 1.33–1.86) per 3.5˚C change. At 10 days post-infection, the probability of an infected mosquito being alive, $A(t)$, was 0.91 (CI: 0.87–0.94) at 21˚C versus 0.67 (CI: 0.58–0.75) at 34˚C, with similar differences for uninfected individuals.

At higher temperatures, fewer mosquitoes develop sporozoites, but those that do, do so faster. We estimate that, between 21˚C and 34˚C, increases in temperature reduced the EIP:

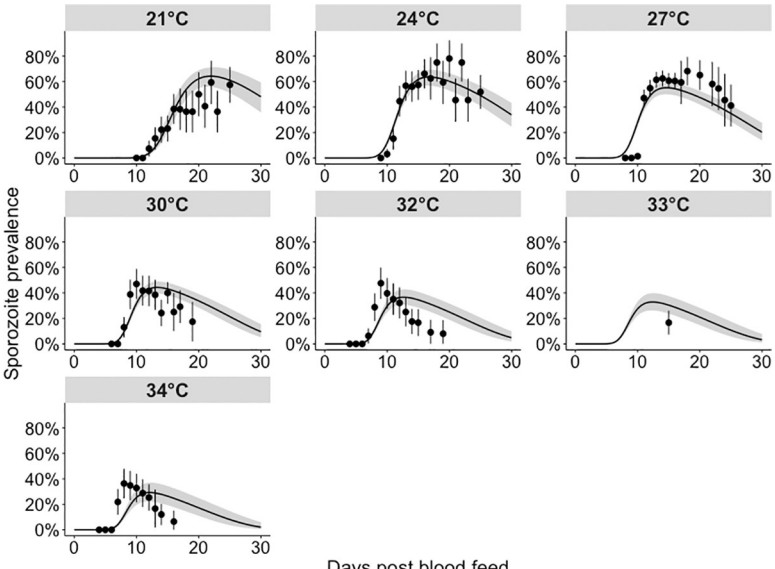

**Fig 3. Model fits to sporozoite prevalence data across all temperatures.** These fits were generated by fitting a single model to all temperatures simultaneously ("*all temperature*" model), with the functional form of temperature as described in Eqs (*2.11*) and (*2.12*). Black points: parasite prevalence of the laboratory mosquito data (95% confidence intervals are given by the vertical black lines). The grey shaded area represents the 95% quantiles of the posterior predictive means; the black lines represent the median posterior predictive means.

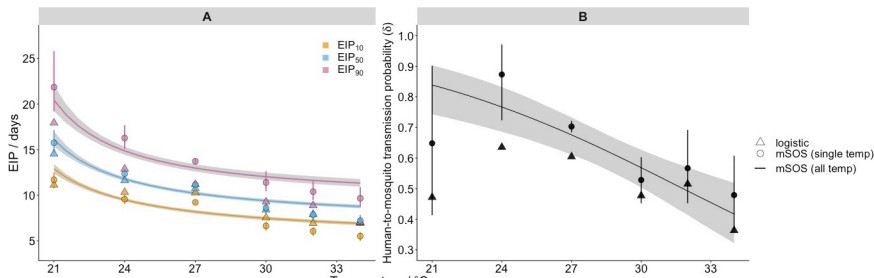

**Fig 4. Effect of temperature on malaria transmission parameters.** Panel A shows the model impact of temperature on the EIP quantiles (as indicated in legend); panel B shows its impact on the human-to-mosquito transmission probability. In both panels, the lines show impact as estimated by the *all temperature* mSOS model with 95% posterior intervals indicated by shading; the discrete round points show the independent estimates from the *single temperature* mSOS model at each temperature, 95% posterior credible intervals are shown by the vertical lines. The discrete triangle points show the logistic growth model parameter estimates at each temperature. The number of iterations used to calculate the EIP plot were thinned to every 5th iteration for efficiency.

$EIP_{10}$ fell from 12.9 days (CI: 12.4–13.4 days) to 6.9 days (CI: 6.7–7.1 days), $EIP_{50}$ declined from 16.1 days (CI: 15.3–17.0) to 8.8 days (CI: 8.6–9.0 days) and $EIP_{90}$ fell from 20.4 days (CI: 19.2–22.1 days) to 11.3 days (CI: 10.9–11.8 days) (Fig 4A). S3 Table provides the EIP percentiles at all temperatures. Increases in temperature reduced the human-to-mosquito transmission probability, which fell from 84% (CI: 74–90%) at 21°C to 42% (CI: 32–52%) at 34°C (Fig 4B).

## Key differences in transmission parameters estimated using mSOS

Next, we compared our estimates of two important malaria transmission parameters–the EIP and human-to-mosquito transmission probability–to those obtained using the predominant literature approach based on logistic regression (logistic model fits are shown S11 Fig). In our framework, we can disentangle the development time of parasites from other underlying processes, specifically mosquito mortality induced by malaria infection, which can influence the observed sporozoite prevalence. In doing so, we quantify the EIP distribution as the time taken for a given percentile of the infected mosquitoes to display sporozoites, in the absence of other mechanisms that determine the observed sporozoite prevalence. This is given by the time at which the cumulative probability an infected mosquito develops any salivary gland sporozoites, $Pr(T_1 \leq t)$, reaches a given value. Using this approach, across all temperatures, the variation in the EIP distribution is greater than the equivalent logistic model estimates (Fig 4A). At 27°C, for example, our *single temperature* model estimated an $EIP_{10}$–$EIP_{90}$ range of 9.2–13.7 days compared to 10.3–11.2 days estimated by the logistic approach. This result was replicated across the different temperatures: by taking into account differences in mosquito survival, our $EIP_{90}$ estimates are higher than those from the logistic method.

The mSOS estimates of the human-to-mosquito transmission probability were consistently higher than the equivalent values estimated by the logistic model (Fig 4B). The 27°C *single temperature* model estimate, for example, was 70% (CI: 68–72%) as opposed to the logistic growth model estimate of 60%. This is because, in the laboratory data we analysed, infected mosquitoes died at an elevated rate, meaning that the proportion of infected mosquitoes declined with time and the observed peak of sporozoite prevalence is not representative of the initial proportion of infected mosquitoes. Not accounting for these mechanisms results in an underestimate of the human-to-mosquito transmission probability.

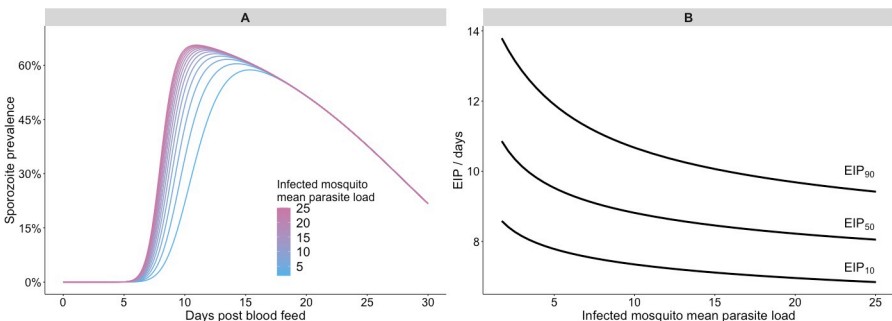

**Fig 5. Modelled impact of parasite load on the extrinsic incubation period.** Panel A shows the impact of varying the mean parasite load of infected mosquitoes on the temporal dynamics of sporozoite prevalence in a sensitivity analysis; panel B summarises how the EIP is affected by the same parameter in the sensitivity analysis. All other parameters were held constant at their mean posterior values.

## Sensitivity analysis of the mean parasite load

Within our model, the time-taken for each simulated parasite to develop is both independent and stochastic. At each time point, whether or not that parasite has yet developed can be viewed as a toss of a coin: the more coins are tossed, the more likely it is that one will land on heads, or, equivalently, a parasite will have developed, by chance. Higher parasite loads then lead to earlier rises in parasite prevalence. Fig 5A shows the simulated sporozoite prevalence over time across populations with different mean parasite load parameter values, which indicates that, within the model, greater parasite numbers cause the prevalence to peak earlier. Fig 5B shows the resultant EIP quantiles as a function of parasite load: increasing the mean parasite load from 1.7 to 25.0 reduced the modelled $EIP_{50}$ from 10.9 to 8.0 days.

## Discussion

Membrane feedings assays form the bulk of experiments used to determine key parasitological parameters of malaria transmission, but despite the seeming simplicity of these experiments, numerous hidden processes contribute to the observed data. Here, we applied a systems biology approach to develop a multiscale model of the temporal dynamics of sporogony (mSOS) that explicitly accounts for these underlying processes. To illustrate its use, we fitted mSOS to experimental data and demonstrated it can produce a reasonable visual fit to the data across a range of experimental protocols. In doing so, we estimated two important determinants of malaria transmission intensity–the EIP and human-to-mosquito transmission probability. By adopting a mechanistic approach, our EIP estimates are defined in terms of the underlying biology rather than characteristics of the raw experimental data. Our estimates indicate greater variation in the EIP than previously thought and highlight the importance of accounting for this variation when making epidemiological predictions (see also [16]). This variation could be included by embedding mSOS within transmission dynamics models of malaria.

The influence of parasite-induced mosquito mortality is still a source of debate and likely to vary depending on the parasite-vector system [27,38]. Nonetheless, accounting for potential differences in mortality is essential to avoid bias in parameter estimates: our estimates of the human-to-mosquito transmission probability, for example, were higher than those estimated by the logistic model. By modelling mosquito infection and survival in a single framework we could estimate malaria infection induced differences in mosquito survival, whilst accounting for mosquitoes that do not develop oocysts within the infectious blood fed group. At higher temperatures, we found that the modelled differences in mosquito survival only partly

explained the sporozoite prevalence data. Another study observed no difference in the survivorship of wild caught *A. gambiae* mosquitoes due to malaria infection, yet *P. falciparum* detection did decline over time [51]. So, it is possible that mosquito survivorship was not completely responsible for these observed declines and alternative explanations, such as sporozoite loss during sugar-feeding [52,53] or sporozoite mortality could require further investigation. Including other such mechanisms in our model could lead to alternative estimates of the EIP. This means that it is possible that the differences between our EIP estimates and those previous could partly be because important, yet unknown, mechanisms were not included in our model.

Temperature is an important predictor of *P. falciparum* prevalence among humans: its impact is, however, often non-linear and location-specific [54–56]. Laboratory data are used to determine temperature-driven variation in the EIP, which is commonly used in the prediction of the spatiotemporal limits and endemicity of malaria transmission [33,57,58]. Over the temperature range investigated, we determined that a linear relationship between temperature and parasite development rate was a reasonable approximation. But, we recognise that the data do not capture the thermal limits of sporogony, and, to handle these would likely require including non-linear relationships (for example see [33]). Our EIP estimates qualitatively match those of [16], which determined that the EIP falls with increases in temperature, and that the incremental change is greatest at lower temperatures. In a semi-field setting, increases in temperature, due to the interaction between deforestation and altitude, decreased the EIP of wild *P. falciparum* [20]. But very little field data exists, and further experimental studies across the whole temperature range of the disease with natural vector-parasite combinations are required to capture the full relationship between temperature and EIP. To more accurately simulate the EIP in the field will require consideration of the impact of the time of biting [59], diurnal temperature fluctuations [60] and mosquito resting location on the temperatures parasites are exposed to during different life stages. The degree to which the EIP of locally adapted strains differ to the laboratory strains must also be considered [61]. Current estimates indicate climate change has the potential to change the number of people at risk of malaria [58,62,63], though there is uncertainty due to the complexity of how the parasite and mosquito will interact with the local changing environment in the long term. Characterising spatiotemporal heterogeneity in the EIP in the field is thus critical to assess our current understanding of malaria and its control as well as the changing risk of malaria resulting from the interactions between climatic, land-use [64] and socio-economic factors [65].

Within our model, heterogeneity in the EIP emerges from differences in the development times of individual parasites. Density-dependent processes [66,67] and intraspecific variation in individual mosquito characteristics, such as body size [68], may also impact the observed sporozoite prevalence. To include such processes in the model in a data-driven way requires higher resolution data to differentiate the impact of different hypotheses. Indeed, to parameterise mSOS, we collated data from four previously published experimental infections that produced data with high temporal resolution. To characterise parasite development, it would be useful if future studies dissected mosquitoes across a greater range of times post infection. It would also be useful if mosquito mortality were recorded: ideally determining the past infection status of carcases through molecular methods.

The work shows a possible mechanism for how higher parasite loads within a mosquito might decrease the EIP, which could potentially have important epidemiological consequences, since it indicates that onwards transmission may be more efficient from more infectious people. This relationship has also recently been considered in another modelling study [36], which found that the impact of parasite load on the EIP of *Plasmodium berghei* is non-linear. Other processes may be operating: emerging experimental evidence hints that a decrease

in resource availability per parasite, as determined by the parasite load and the number of times the mosquito blood feeds, may decrease the parasite development rate [50,69]. So, to fully capture this relationship may require the incorporation of density-dependent mechanisms. Since (a) transmission is highly sensitive to changes in EIP, (b) transmission blocking interventions cause a decline in oocyst intensity [40], and (c) parasite load in the field may be higher than previously thought [39], the influence of parasite load on EIP merits further investigation.

It is now well over a century since *Plasmodium* parasites were first uncovered in dissected anopheline mosquitoes. Today, we remain dependent on dissection for understanding the parasite lifecycle in mosquitoes, and there remains much to learn. The model we introduce here provides a new way to parse dissection data to probe the underlying biology. Our model, or ones like it, could be extended to incorporate more fine scale characteristics of parasite ecology, but to do so in a principled manner requires more fine scale data. As such, we foresee a great necessity and opportunity for closer collaboration between experimentalists and modellers in the future.

## Supporting information

**S1 Code. Mathematica file that demonstrates the derivation of the model.**
(NB)

**S1 Text. Supporting Information: survival analysis and logistic growth model methods.**
(DOCX)

**S1 Table. Model priors.**
(DOCX)

**S2 Table. All temperature model posterior values.**
(XLSX)

**S3 Table. EIP estimates by temperature.** The posterior predictive median, 5% and 95% quantiles of the all temperature model EIP estimates are given to one decimal place.
(XLSX)

**S1 Fig. Single temperature model estimates of parasite transition parameters.** Panels show the posterior estimates of the two parameters (A: shape; B: rate) of the gamma distribution governing the development time between inoculation and observed sporozoites for the single temperature models. The posterior median and difference between the 2.5 and 97.5 posterior quantiles are represented by the points and vertical lines respectively.
(TIF)

**S2 Fig. Kaplan-Meier estimates of the mosquito survival data used in the model fitting.** Mosquito survival data was obtained from three previously published studies [14,19,38]. Mosquitoes fed on infectious blood are shown in blue, mosquitoes fed on uninfected (control) blood are shown in yellow.
(TIF)

**S3 Fig. Trace plots of the MCMC parameter sampling for the all temperature model.** The horizontal and vertical axis give the iteration and parameter values respectively. Iterations during the warmup are not included.
(TIF)

**S4 Fig. Pairwise comparison of the MCMC parameter samples for the all temperature model.** Parameter syntax is as follows: $\alpha_{GO}$ ("shape_oocyst"), $m_\beta$ ("m_rate_oocyst"), $c_\beta$

("c_rate_oocyst"), $\alpha_{OS}$ ("shape_sporozoite"), $\beta_{OS}$ ("rate_sporozoite"), $\mu$ ("mu_NB"), k
("k_NB"), $m_\delta$ ("m_delta"), $c_\delta$ ("c_delta"), a ("a"), b ("b"), $\beta_E$ ("beta_inf"), $\beta_C$ ("beta_temp"), $\sigma_\delta$
("sigma_error_delta") and $\sigma_{survival}$ ("sigma_error_survival").
(TIF)

**S5 Fig. Posterior and prior probability densities.** The posterior distributions are estimated
by kernel density estimation with a gaussian smoothing kernel.
(TIF)

**S6 Fig. Modelled times required for individual malaria parasites to transition between life
stages.** Panels A and B show the probability density function (PDF) and the cumulative probability density function (CDF) that an individual *P. falciparum* parasite within a mosquito
(maintained under standard insectary conditions: 27˚C) will transition from a given life stage
to the next life stage at a given time post blood feed. The grey shaded area represents 2.5%-
97.5% posterior quantiles of the estimated distributions.
(TIF)

**S7 Fig. Single temperature model fits of the temporal variation in sporozoite prevalence.**
Black points: parasite prevalence of the laboratory mosquito data (95% binomial confidence
intervals are given by the vertical black lines). The grey shaded area represents the 95% uncertainty intervals of the mean prevalence (posterior predictive means). The black line represents
the median of the poster predictive means.
(TIF)

**S8 Fig. Kaplan-Meier survival curves and all temperature Cox proportional hazards survival model fit.** Kaplan Meier curves are stepped. The shaded area shows the 95% uncertainty
intervals of the posterior predictive mean survival probability (A(t)) modelled by the Cox proportional hazards model.
(TIF)

**S9 Fig. Kaplan Meier survival curves and the single temperature Cox proportional hazards
survival model fits.** Kaplan Meier curves are stepped. The shaded area shows the 95% uncertainty intervals of the posterior predictive mean survival probability (A(t)) modelled by the
Cox proportional hazards model.
(TIF)

**S10 Fig. All temperature model fits of the temporal variation in oocyst prevalence and
oocyst intensity.** Oocyst prevalence plot: black points represent the prevalence of the laboratory mosquito data and the lines represent the 95% binomial confidence intervals. The grey
shaded area represents the 95% uncertainty interval of the posterior predictive mean prevalence. Oocyst intensity plots: black points represent the mean oocyst intensity among all blood
fed mosquitoes, the black line shows the posterior predictive mean oocyst count, and the grey
shaded area represents the uncertainty in the mean oocyst count indicating the 2.5% and
97.5% negative binomial quantiles when fixing all parameters at their mean posterior values.
(TIF)

**S11 Fig. Logistic growth model fits.** Logistic model fits (black line) to a subset of the sporozoite prevalence data (i.e. all data before observed peak sporozoite prevalence) are shown as solid
lines; black points indicate those points were included in the fitting; grey points show those
excluded.
(TIF)

## Acknowledgments

We would like to thank Professor Matthew Thomas for useful insight and the Thomas lab at Penn State University for producing and publishing the high-quality data upon which our study is based.

## Author Contributions

**Conceptualization:** Isaac J. Stopard, Thomas S. Churcher, Ben Lambert.

**Data curation:** Isaac J. Stopard, Thomas S. Churcher.

**Formal analysis:** Isaac J. Stopard, Thomas S. Churcher, Ben Lambert.

**Funding acquisition:** Isaac J. Stopard, Thomas S. Churcher.

**Investigation:** Isaac J. Stopard, Thomas S. Churcher, Ben Lambert.

**Methodology:** Isaac J. Stopard, Thomas S. Churcher, Ben Lambert.

**Supervision:** Thomas S. Churcher, Ben Lambert.

**Visualization:** Isaac J. Stopard, Thomas S. Churcher, Ben Lambert.

**Writing – original draft:** Isaac J. Stopard, Thomas S. Churcher, Ben Lambert.

**Writing – review & editing:** Isaac J. Stopard, Thomas S. Churcher, Ben Lambert.

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
