## [Decision Letter · Decision Letter 0]

9 Nov 2020

Dear Mr Stopard,

Thank you very much for submitting your manuscript "Estimating the extrinsic incubation period of malaria using a mechanistic model of sporogony" for consideration at PLOS Computational Biology.

As with all papers reviewed by the journal, your manuscript was reviewed by members of the editorial board and by several independent reviewers. In light of the reviews (below this email), we would like to invite the resubmission of a significantly-revised version that takes into account the reviewers' comments.

We cannot make any decision about publication until we have seen the revised manuscript and your response to the reviewers' comments. Your revised manuscript is also likely to be sent to reviewers for further evaluation.

Sincerely,

Miles P. Davenport, MB BS, D.Phil

Associate Editor

PLOS Computational Biology

Nina Fefferman

Deputy Editor

PLOS Computational Biology

Reviewer's Responses to Questions

**Comments to the Authors:**

Reviewer #1: In their paper, Stopard et al. model the time required for sporogony using a multiscale stochastic model and fit their model to experimental data from different studies. They compare their results to results obtained by using an existing model and find a greater variation in the extrinsic incubation period (EIP) using their model. The EIP is the duration of sporogony from uptake of parasites until infectious sporozoites can be found in the mosquito salivary glands. Overall, this reviewer thinks that this study addresses an important part of the malaria parasite life cycle, they use a model with greater flexibility than the predominantly (according to the authors) used logistic model, and their model has the potential to improve our understanding of the mosquito stage of malaria. However, some aspects of the modelling require further clarification.

This reviewer’s expertise is not in the mosquito stage of malaria, thus it is difficult to assess the validity of Stopard et al.’s model assumptions or the significance and impact of their findings on the field of malaria parasite biology in mosquitos. However, regarding the mathematical model, the authors present a novel multiscale model of this biological process. At the parasite scale, they model the time until the parasite is in the next development stage as following a gamma distribution. At the mosquito scale, they take into account the human-to-mosquito transmission probability and death of mosquitos and at the observational level they model the fraction of infected mosquitos and the number of oocysts found in dissected mosquitos. The main results are the temporal dynamics of sporogony, the impact of temperature on the EIP from the model and the comparison of their model with a logistic model. Stopard et al. find a greater variation in the EIP than with a logistic model and a higher human-to-mosquito transmission probability. They argue that since estimates of transmission probability are highly sensitive to changes in EIP, a better understanding of the EIP is necessary to improve estimates of malaria transmission. Their methods appear to be technically sound and the authors clearly state their model assumptions and justify them. Nonetheless, questions regarding the logistic model, the comparison of the two models, and the what informs the heterogeneity in their model remain.

Major comments:

1. Some aspects regarding the logistic model and the comparison of the two models are not clear to this reviewer. For example, why was a part of the data excluded? What were the criteria for excluding later observations? It appears that all data after the highest mean sporozoite prevalence was reached were excluded. Given that the logistic model does not appear to include mosquito death, the higher human-to-mosquito transmission probability of Stopard et al.’s model does not seem surprising. To what extent may the difference in the variation in the EIP be due to the different model types (see also comment 2)?

2. What informs the heterogeneity found by the model? Is it heterogeneity in the data, the construction of the model that allows for distributions e.g. in parasite development times, or is it due to the choice of the priors? If it is informed by heterogeneity in the data, then in this reviewer’s opinion the manuscript could be improved by including a table or figure (maybe in the supplement) that shows the heterogeneity in the data and compares it to the heterogeneity in the two models. In this case, it might also be beneficial to show (e.g. in a supplementary figure) that the heterogeneity in their model does not depend on the choice of the priors but can also be found using uninformative priors instead of weakly informative priors.

3. In the discussion (lines 516-517), the authors claim to have shown “predictive accuracy”. In this reviewer’s opinion they have shown that their model is a visually better fit to the data than a logistic model, which has less flexibility and was fit to only a part of the data. It is not clear in which way they have shown “predictive accuracy”.

Minor comments:

4. Equation (2.08) (lines 296-299), should E(.) be replaced by En(.)?

5. The authors state their model is mechanistic, but it seems to this reviewer that it is describing features of the biological process (e.g., the time for the parasite to reach the next development stage) rather than mechanisms causing these features.

Reviewer #2: Stopard et al. developed a mechanistic mathematical model of malaria parasite sporogony and estimated the parameters of the model by fitting the model to a multitude of experimental data. They found their mechanistic model predicted different EIPs from previously reported values and what a statistical model (e.g. logistic model) predicted. In addition, they also examined the impact of temperature on EIP.

Controlling malaria transmission from human to mosquito represents one of the crucial approaches to achieving malaria elimination. Given that a quantitative understanding of how different biological factors influence the probability of successful transmission is still very limited, this study represents a very good effort to fill the knowledge gap using data-based modelling approaches. The manuscript is well written, and the models and methods are proposed rigorously and presented clearly. Therefore, I think this work will be suitable for PLoS Computational Biology after addressing the following comments.

1. Lines 177-180: provide some rationales for the assumptions of which are time-dependent and which are not.

2. For model fitting, are there any correlations between model parameters identified based on posterior samples? if there are, how do they influence the parameter estimates reported in the main text?

I also notice that some priors (in Table S1) are Gaussian with an SD of 2.5, e.g. alpha_OS, alpha_GO, mu, k, a and b, etc. It is fine for mu, k, a and b because the priors look less informative compared to the posteriors. However, for alpha_OS and alpha_GO, the priors may be very influential. So could the authors provide some justifications for the choice of SD = 2.5 as “prior knowledge”? If 2.5 was chosen arbitrarily, the authors should relax the constraint, say using a larger SD, and examine if the estimates are robust to the choice of the SD.

3. When comparing the waiting time before sporozoite positive between model prediction and experimental observation, i.e. lines 402-404, it would be more informative to report a comparison at a comparable prevalence such that the model prediction is more relevant to the “reailty”. For example, instead of only examining EIP10 or higher, predictions of EIP1, EIP2 and EIP5 should also be included in the results.

Minor comments:

Line 23: EIP10 and EIP90 (and EIP50) were not defined in the abstract.

Line 374: provide Stan version number

Reviewer #3: This manuscript by Stopard and colleagues describes the development of a novel within-vector mechanistic model of a key step in the transmission process of the malaria parasite. The analysis is thorough, well described and the writing of the manuscript is exceptional throughout. While I do have some specific queries about some parts of the analysis and how it is presented, overall I believe the stud is of high quality and should be published.

Major comments:

This study fits the model to data from four pre-existing studies. It is not clear in this manuscript why these four studies were chosen. Presumably there are many more studies that have measured EIP previously. Classically I would have expected either a thorough systematic literature search to identify all possible data sources or a strong justification of why a particular study / small group of studies was chosen for this analysis, but currently there is neither.

While the model has clear strengths for representing and predicting various transmission processes, I did find the suitability of the model to represent changes in response to temperature changes less convincing compared to other results. The data used span a limited range of temperatures (particularly excluding extreme low or high values), a step of the analysis fixes a linear development rate (line 346) based on the analysis of data from a limited range of temperatures and the fit of the model shows increasing disparity in shape with higher temperature (Figure 3). These are fairly minor issues in themselves, but in the discussion mSOS is explicitly compared to models designed to explore the thermal limits of transmission (line 541). While the results are caveated and some development is suggested, I just don’t see this particular application of mSOS as having been conclusively proven in this manuscript.

Minor comments:

Approximation of T(GS) (line 218) – is this just for studies that didn’t observe intermediate parasite forms? Worth stating if so, also any idea how good this approximation is?

Study random effect- defining unique studies as different study/ temperature combinations seems strange when you are also trying to estimate the effect of temperature alone. Might be better to restrict this to just unique study combinations as I would expect between study variation to be much greater than within study but at different temperature variation.

**Have all data underlying the figures and results presented in the manuscript been provided?**

Reviewer #1: None

Reviewer #2: Yes

Reviewer #3: Yes

PLOS authors have the option to publish the peer review history of their article (what does this mean?). If published, this will include your full peer review and any attached files.

Reviewer #1: No

Reviewer #2: No

Reviewer #3: No
---

## [Decision Letter · Decision Letter 1]

28 Dec 2020

Dear Mr Stopard,

We are pleased to inform you that your manuscript 'Estimating the extrinsic incubation period of malaria using a mechanistic model of sporogony' has been provisionally accepted for publication in PLOS Computational Biology.

Best regards,

Miles P. Davenport, MB BS, D.Phil

Associate Editor

PLOS Computational Biology

Nina Fefferman

Deputy Editor

PLOS Computational Biology

Reviewer's Responses to Questions

**Comments to the Authors:**

Reviewer #1: In the original paper submitted by Stopard et al., this reviewer’s major comments mainly concerned the logistic model, its comparison with the author’s model “mSOS”, and the heterogeneity in the model. Regarding the logistic model, Stopard et al. now clearly state which data was excluded and that it was excluded to follow a previously used method in order to compare it to their model. In the revised version and their response, Stopard et al. have given more details on the differences between the logistic model and mSOS and how including mechanisms such as mosquito mortality avoids bias in parameter estimates. Regarding the heterogeneity in the model, Stopard et al. responded that the heterogeneity of the model is due to heterogeneity in the data and also the model choice.

In this reviewer’s opinion, the manuscript is well-written and studies and improves modelling of an important part of the malaria parasite life cycle. All previous comments were sufficiently addressed by the authors and there are no additional comments. This reviewer’s recommendation is that this manuscript is now appropriate for publication in PLOS Computational Biology.

Reviewer #2: The authors have addressed all my concerns.

Reviewer #3: The authors have sufficiently addressed all of my comments

**Have all data underlying the figures and results presented in the manuscript been provided?**

Reviewer #1: Yes

Reviewer #2: Yes

Reviewer #3: Yes

PLOS authors have the option to publish the peer review history of their article (what does this mean?). If published, this will include your full peer review and any attached files.

Reviewer #1: No

Reviewer #2: No

Reviewer #3: **Yes: **Oliver Brady

---

## [Editor Report · Acceptance letter]

7 Feb 2021

PCOMPBIOL-D-20-01746R1 

Estimating the extrinsic incubation period of malaria using a mechanistic model of sporogony

Dear Dr Stopard,

I am pleased to inform you that your manuscript has been formally accepted for publication in PLOS Computational Biology. Your manuscript is now with our production department and you will be notified of the publication date in due course.

With kind regards,

Alice Ellingham
